# Angiopoietin-2 in white adipose tissue improves metabolic homeostasis through enhanced angiogenesis

Yu A An[1†], Kai Sun[1,2†], Nolwenn Joffin[1], Fang Zhang[1,3], Yingfeng Deng[1], Olivier Donzé[4], Christine M Kusminski[1], Philipp E Scherer[1,5*]

[1]Touchstone Diabetes Center, Department of Internal Medicine, The University of Texas Southwestern Medical Center, Dallas, United States; [2]Center for Metabolic and Degenerative Diseases, The Brown Foundation Institute of Molecular Medicine, University of Texas Health Science Center at Houston, Houston, United States; [3]Institute for Nutritional Sciences, Shanghai Institutes for Biological Sciences, Chinese Academy of Sciences, Shanghai, China; [4]AdipoGen Life Sciences, Epalinges, Switzerland; [5]Department of Cell Biology, The University of Texas Southwestern Medical Center, Dallas, United States

*For correspondence: philipp.scherer@utsouthwestern.edu

†These authors contributed equally to this work

Competing interests: The authors declare that no competing interests exist.

**Abstract** Despite many angiogenic factors playing crucial roles in metabolic homeostasis, effects of angiopoietin-2 (ANG-2) in adipose tissue (AT) remain unclear. Utilizing a doxycycline-inducible AT-specific ANG-2 overexpression mouse model, we assessed the effects of ANG-2 in AT expansion upon a high-fat diet (HFD) challenge. ANG-2 is significantly induced, with subcutaneous white AT (sWAT) displaying the highest ANG-2 expression. ANG-2 overexpressing mice show increased sWAT vascularization and are resistant to HFD-induced obesity. In addition, improved glucose and lipid metabolism are observed. Mechanistically, the sWAT displays a healthier expansion pattern with increased anti-inflammatory macrophage infiltration. Conversely, ANG-2 neutralization in HFD-challenged wild-type mice shows reduced vascularization in sWAT, associated with impaired glucose tolerance and lipid clearance. Blocking ANG-2 causes significant pro-inflammatory and pro-fibrotic changes, hallmarks of an unhealthy AT expansion. In contrast to other pro-angiogenic factors, such as vascular endothelial growth factor-A (VEGF-A), this is achieved without any enhanced beiging of white AT.

## Introduction

Caloric overload leading to obesity has assumed epidemic proportions. Obesity is frequently associated with metabolic dysfunction, such as type 2 diabetes, cardiovascular diseases and cancer. However, short of bariatric surgery or life style improvements, our therapeutic options are limited (*Kusminski et al., 2016*). Therefore, a better understanding of the mechanistic underpinnings of the pathological events associated with obesity may define effective targets for pharmacological interventions.

Adipose tissue (AT) is the most dynamic and plastic organ in adults. Upon exposure to different metabolic challenges, AT has the capacity to either expand or shrink according to nutrient status. Meanwhile, blood vessel growth and regression must also occur simultaneously (*Sun et al., 2011*). Given the essential contributions of blood vessels to AT homeostasis (*Cao, 2010*, *2013*), the process of angiogenesis (i.e. new blood vessel formation) is a key factor for healthy AT expansion. Notably, there are several pro-angiogenic factors secreted by adipocytes, such as leptin, adiponectin, vascular endothelial growth factor-A (VEGF-A), VEGF-B and angiopoietins (*Cao et al., 2001*; *Lee et al.,*

*2015*; *Shibata et al., 2004*; *Sun et al., 2012*; *Michailidou et al., 2012*; *Robciuc et al., 2016*; *Voros et al., 2005*; *Xue et al., 2008*). This reflects the ability to self-regulate angiogenesis in AT. We have previously shown that angiogenesis is the rate-limiting step for healthy AT expansion (*Sun et al., 2012*). Enhanced angiogenesis facilitates healthy AT expansion, associated with enhanced adipogenesis, decreased AT inflammation and minimized fibrotic damage. In contrast, inadequate angiogenesis leads to AT hypoxia, enlarged adipocyte size, increased inflammation and fibrosis, representing an unhealthy pattern in AT expansion (*Sun et al., 2011*). Promoting angiogenesis during early AT expansion, therefore, has beneficial effects on AT function and systemic metabolic homeostasis (*Sun et al., 2012*). We reported earlier that local overexpression of VEGF-A in white AT (WAT) and brown AT (BAT) both results in improved vascularization and resistance to high-fat diet (HFD)-induced metabolic dysfunction (*Sun et al., 2012*, *2014*). A recent study also reported that VEGF-B can lead to enhanced AT vasculature, thereby exerting beneficial metabolic effects (*Robciuc et al., 2016*). Nevertheless, for a better understanding of the process, additional angiogenic factors still need to be examined in the context of AT expansion.

Aside of the family of VEGF factors, the angiopoietin (ANG) family is also involved in vascular remodeling, maturation and stabilization (*Thomas and Augustin, 2009*) and includes mainly ANG-1 and ANG-2. Although the role of ANGs in tumor angiogenesis has been well studied, their functions in AT are poorly defined. There is evidence suggesting that in *ob/ob* mice, ANG-1 expression is decreased in AT, while ANG-2 expression is increased (*Voros et al., 2005*). Other data suggest that ANG-2 induced by FOXC2 overexpression results in enhanced blood vessel sprouting in white AT (*Xue et al., 2008*). Our own data show that under different metabolic conditions, such as exercise, cold exposure, HFD feeding and fasting, *Angpt2* mRNA levels in the ATs are also subject to change (*Figure 1—figure supplement 1A–1E*). The physiological regulation of ANG-2, therefore, further supports the idea that ANG-2 plays an important functional role in AT physiology. There is an ongoing debate whether ANG-2 leads to vascular sprouting or regression, depending on the tissue under study. However, the specific contributions of ANG-2 to AT are unknown. Therefore, we utilized a genetic approach with a mouse model that specifically overexpresses ANG-2 in AT to help us define the effects of ANG-2 in AT angiogenesis and AT expansion upon exposing mice to a metabolic challenge, such as HFD.

Here, we report results from a doxycycline (Dox)-inducible AT-specific ANG-2 overexpression mouse model with substantial induction of ANG-2 in sWAT. ANG-2 overexpressing mice show increased vascularization and reduced inflammatory changes in sWAT, leading to healthy AT expansion. This results in resistance to HFD-induced weight gain and improvements in metabolic function, including enhanced glucose tolerance, insulin sensitivity and lipid disposal. Conversely, we also demonstrate that antagonizing endogenous ANG-2 by neutralizing antibodies has opposite effects. Blocking ANG-2 causes a significant decrease in vascular density in sWAT, associated with an unhealthy expansion of AT defined by a highly pro-inflammatory microenvironment and pro-fibrotic changes. ANG-2 neutralization leads to an exacerbation of HFD-induced metabolic changes. Both gain- and loss-of-function studies of ANG-2 in AT therefore highlight the important role this factor plays in AT physiology.

## Results

### Inducible overexpression of ANG-2 in WAT promotes new blood vessel formation

To directly evaluate the role of ANG-2 in AT, we took advantage of a mouse model carrying a tetracycline responsive element (TRE)-driven ANG-2 transgenic cassette (TRE-ANG2), shown schematically in *Figure 1A*. To obtain adipocyte-specific overexpression, we used a reverse tetracycline-dependent transcriptional activator (rtTA) which is driven by the adiponectin promoter (Adipo-rtTA). rtTA can bind and activate the TRE in the presence of Dox. Thus, we obtained the double transgenic mouse model (Adipo-ANG2) by crossing the TRE-ANG2 mice to the Adipo-rtTA mice and achieved an inducible and tissue-specific overexpression model of ANG-2 in the AT. To confirm both inducibility and tissue-specificity, we fed the mice with a chow diet containing 600 mg/kg Dox. After 5 weeks of Dox feeding, we harvested different fat pads and other tissues from both Adipo-ANG2 and Adipo-rtTA mice (serving as controls) and assessed *Angpt2* mRNA levels. While both subcutaneous

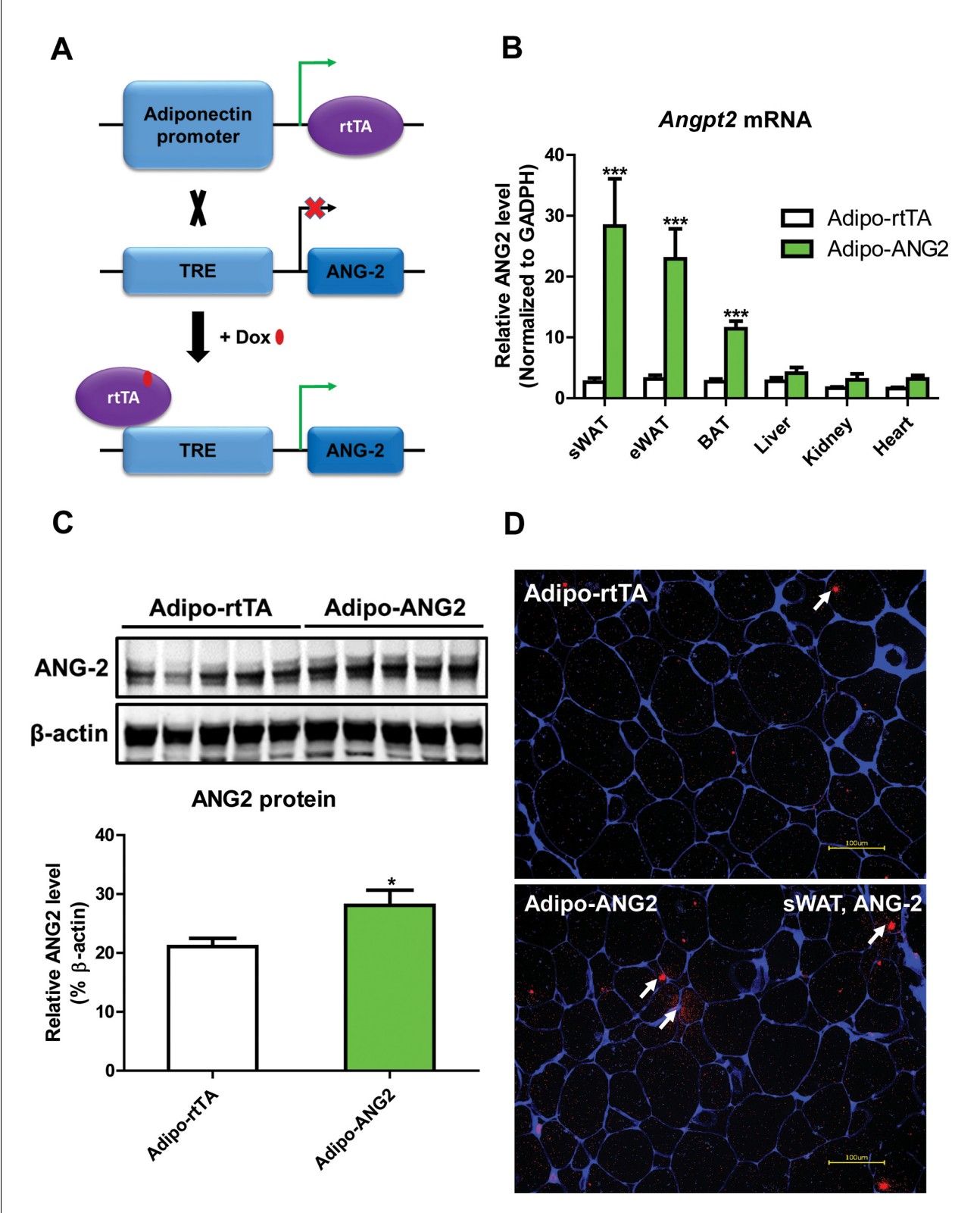

**Figure 1.** Adipocyte-specific, Dox-inducible overexpression of ANG-2. (**A**) Schematic illustration of the adipocyte-specific, Dox-inducible ANG-2 overexpression mouse model. The adiponectin promoter-driven reverse tetracycline-dependent transcriptional activator (rtTA) mice were bred to tetracycline responsive element (TRE)-driven ANG-2 (TRE-ANG2) to generate the double transgenic mouse model. rtTA activity is activated to induce ANG-2 overexpression in the presence of Dox. (**B**) *Angpt2* transcriptional changes in different tissues from control (Adipo-rtTA) mice and

*Figure 1 continued on next page*

Figure 1 continued

overexpressing (Adipo-ANG2) mice, fed with HFD/Dox 600 mg/kg for five weeks. (C) Representative Western blot image (upper panel) and quantitative analysis (lower panel) of ANG-2 and β-actin protein levels in sWAT from HFD/Dox 600 mg/kg fed Adipo-rtTA mice and Adipo-ANG2 mice. (D) Representative images of ANG-2 immunofluorescence staining (Red, Magnification = 200 × ) in both Adipo-rtTA control and Adipo-ANG2 double transgenic groups after Dox induction. Perilipin (Blue) was also stained to label adipocytes. Bar = 100 μm. For all the statistical graphs: n = 5 ~ 7 mice per group. Data are shown as mean ± s.e.m. *p<0.05.
The following figure supplement is available for figure 1:

**Figure supplement 1.** Differential regulation of *Angpt2* mRNA under variety of metabolic challenges.

and epididymal WAT (sWAT and eWAT, respectively) show a significant increase in *Angpt2* transcription, there was no elevation of *Angpt2* mRNA levels in the liver, kidney or heart (*Figure 1B*). Notably, we also noticed enhancement of *Angpt2* transcription in brown adipose tissue (BAT). Since the most abundant expression of *Angpt2* mRNA was observed in sWAT, we examined the protein levels of ANG-2 in sWAT by Western blotting. Compared to Adipo-rtTA control mice, ANG-2 protein levels are significantly elevated in Adipo-ANG2 transgenic mice (*Figure 1C*). In addition, immunofluorescence staining of ANG-2 in sWAT reveals a substantially higher ANG-2 signal in Adipo-ANG2 mice (*Figure 1D*).

Upon confirming the enhanced expression of ANG-2 in AT, we moved forward to address the question whether ANG-2 either promotes or impedes AT vascularization upon a HFD challenge. After the mice were subjected to HFD/Dox containing diet for five weeks, a number of endothelial and angiogenic markers were assessed by qPCR in sWAT of both Adipo-rtTA control mice and Adipo-ANG2 mice. *Vegfa*, the angiopoietin receptor TIE2 (gene name *Tek*), the endothelial marker CD31 (*Pecam1* as the gene name) and the angiogenesis marker tumor endothelial marker-8 (TEM8, *Antxr1*) are all significantly up-regulated in Adipo-ANG2 transgenic mice, indicating that the local angiogenic program is activated upon ANG-2 induction (*Figure 2A*). In addition, the vascular density of sWAT was determined by immunostaining with antibodies to an additional endothelial marker, endomucin (*Liu et al., 2001*) (*Figure 2B*) with a quantitative analysis shown in *Figure 2C*. This reveals dramatically increased vascular density in Adipo-ANG2 transgenic mice. It is also important to assess the relative level of hypoxia in sWAT upon increasing the vasculature after ANG-2 overexpression. We observe a significant decrease in hypoxia inducible factor-1α (HIF-1α) protein and gene expression levels, as well as a significant reduction in the expression levels of the HIF-1α downstream target gene *Slc2a1* (gene symbol for GLUT1) in ANG-2 transgenic mice (*Figure 2—figure supplement 1A–C*). In parallel, we also performed pimonidazole staining in sWAT of ANG-2 transgenic mice, to further validate that ANG-2 overexpression dramatically reduces AT hypoxia, as judged by decreased pimonidazole-positive signal (*Figure 2—figure supplement 1D*). Thus, elevated local levels of ANG-2 specifically in WAT potently promote a functional angiogenic program.

## ANG-2 transgenic mice show improved metabolic responses upon a HFD challenge

The increased levels of ANG-2 result in an enhanced vascular density in sWAT. We wanted to probe the consequences of this pro-angiogenic phenomenon on whole body metabolism. Both the Adipo-ANG2 and Adipo-rtTA mice were challenged with a HFD/Dox diet for five weeks. During this period, body weights were continuously monitored. Adipo-ANG2 transgenic mice weigh less than control mice (*Figure 3A*). The reduced body weight is due to a significant decrease in fat mass and increased lean mass (*Figure 3B*). Oral glucose tolerance tests (OGTTs) reveal that ANG-2 overexpression significantly improves glucose clearance (*Figure 3C*). Improved insulin sensitivity is also observed in the Adipo-ANG2 transgenic mice, as judged by insulin tolerance tests (ITT) (*Figure 3D*). Consistent with the ITT results, we also demonstrate that ANG-2 overexpression leads to an improvement of insulin signaling in sWAT, reflected by an increased level of phosphorylated Akt upon insulin stimulation (*Figure 3—figure supplement 1*). Interestingly, we also observe a similar improvement of insulin signaling in the liver (data not shown). Enhanced angiogenesis in AT can lead to increased energy expenditure (*Sun et al., 2014*). We therefore assessed this possibility by analyzing cohorts in metabolic cages. Both oxygen consumption (VO$_2$) and carbon dioxide generation

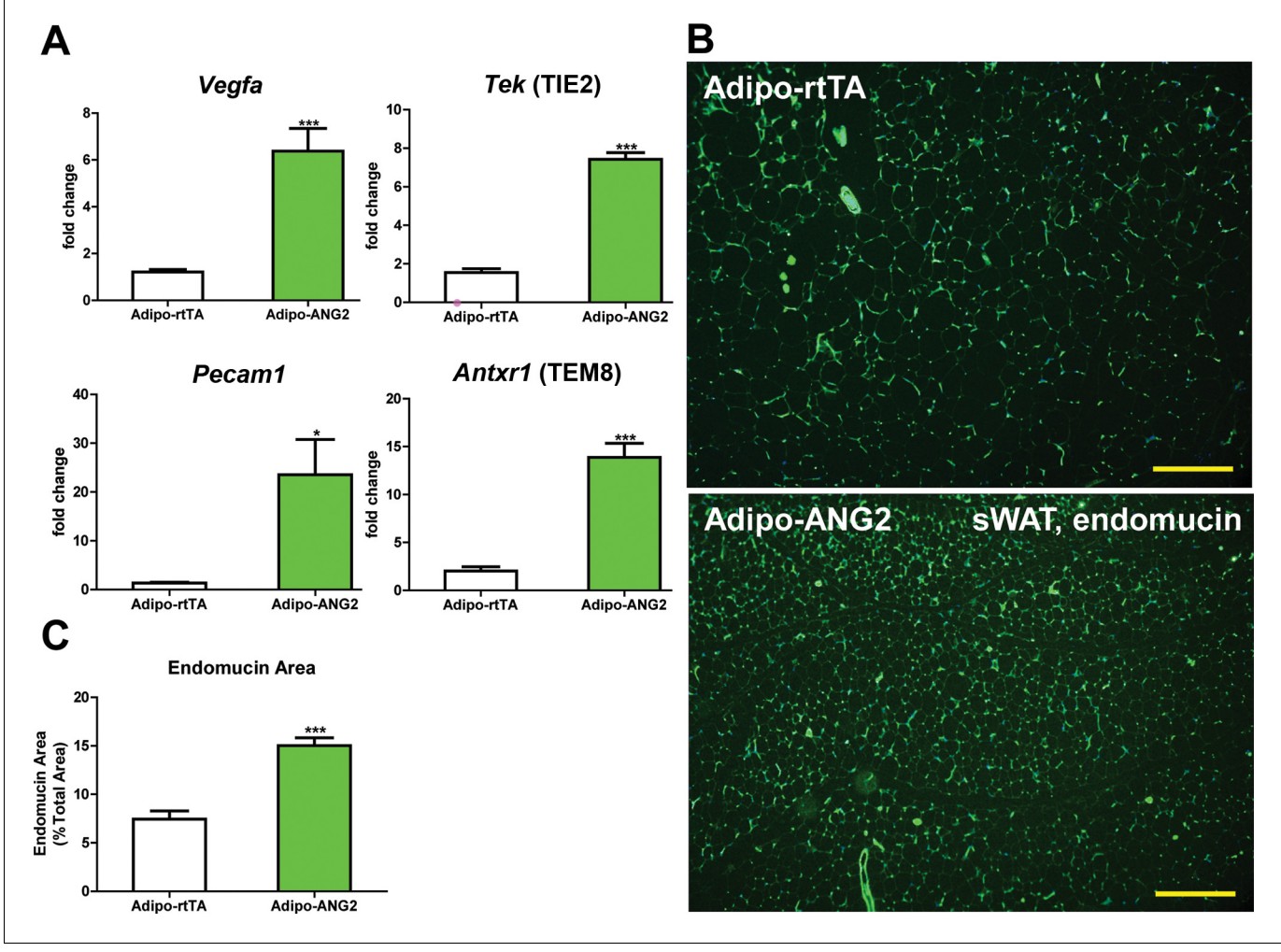

**Figure 2.** Vascular changes after overexpression of ANG-2. (**A**) Angiogenic factors and endothelial cell markers including *Vegfa, Tek, Pecam1* and *Antxr1* mRNA levels in the harvested sWAT from 5 week HFD/Dox 600 mg/kg induced Adipo-rtTA and Adipo-ANG2 mice.(**B**) Representative immunofluorescence staining image of endothelial marker endomucin (Green, Magnification = 100×) and (**C**) the statistical result of relative endomucin area calculated by the percentage of endomucin positive signals in the total area. SWAT section staining images from both groups are shown. Bar = 160 µm. For all the statistical graphs: n = 6 mice per group. Data are shown as mean ± s.e.m. *p<0.05, ***p<0.001.

The following figure supplement is available for figure 2:

**Figure supplement 1.** Hypoxic parameters after ANG-2 overexpression in sWAT.

($VCO_2$) for the ANG-2 transgenic mice were significantly enhanced during light and dark cycles, reflecting higher energy expenditure upon ANG-2 overexpression (*Figure 3E*). However, there is no change in the respiratory exchange ratio (RER) between genotypes (data not shown).

Beyond the effects on carbohydrates, Adipo-ANG2 transgenic mice also show improvements in lipid metabolism. The serum lipid profile (*Figure 3F*) shows that both non-esterified free fatty acids (NEFAs) and cholesterol levels were significantly decreased in Adipo-ANG2 mice, while serum triglycerides only showed a trend towards a decrease. Upon further evaluation with an oral lipid gavage, it is apparent that Adipo-ANG2 transgenic mice display significantly higher efficiency in lipid clearance (*Figure 3G*). These improvements in lipid metabolism are in part due to a significantly enhanced lipoprotein lipase (LPL) activity, both systemically and locally (*Figure 3H*). Furthermore, we also detected the expression of angiopoietin-like proteins (ANGPTLs), a family of proteins sharing similar structures with ANGs. Several of the family members exert inhibitory effects on lipoprotein

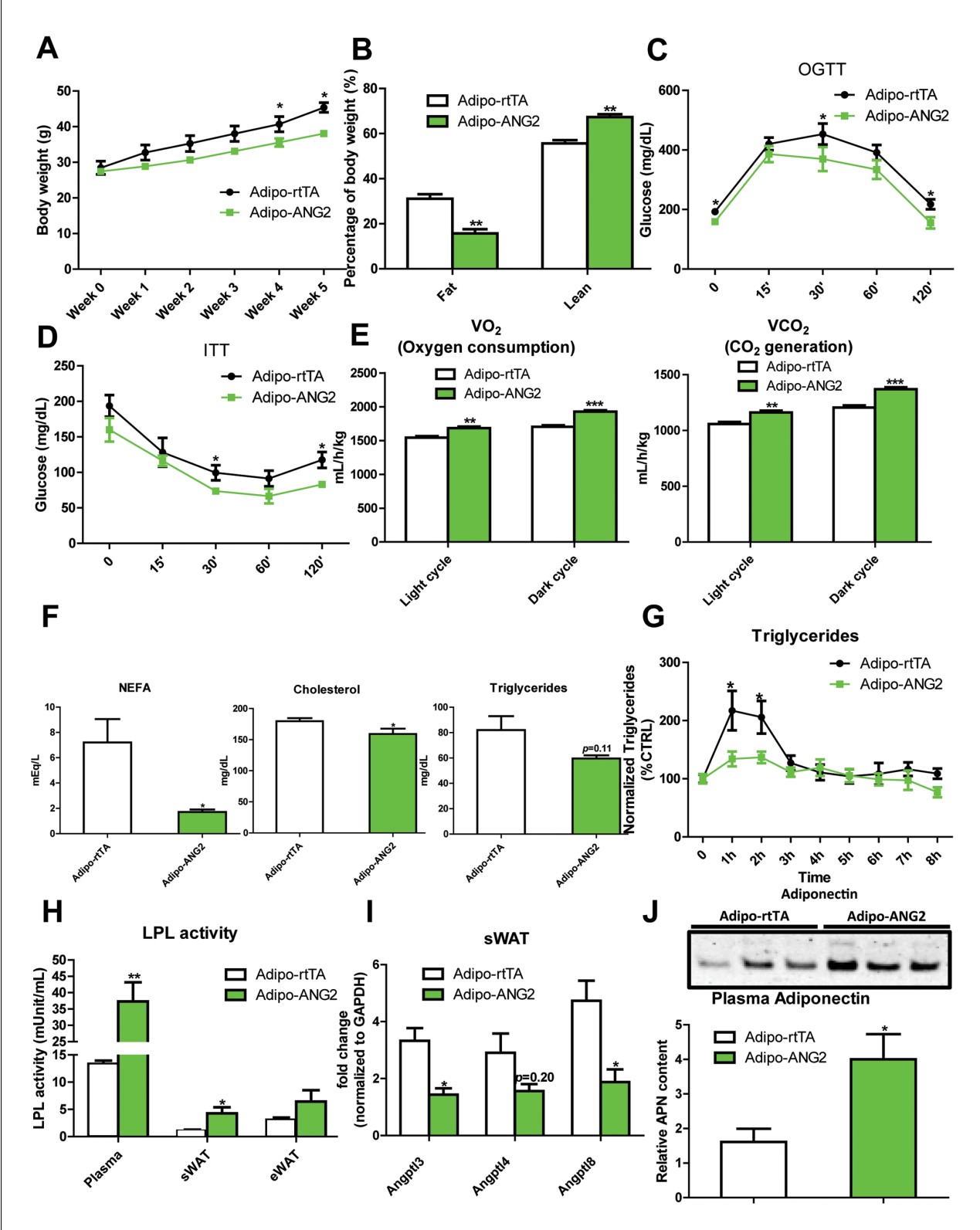

**Figure 3.** Metabolic effects in mice overexpressing ANG-2. After HFD/Dox 600 mg/kg induction for five weeks, Adipo-rtTA control and Adipo-ANG2 transgenic mice were subjected to the following metabolic analyses: (A) Body weight was continuously monitored for five weeks.(B) Body composition data obtained via MRI. Both the percentage of fat mass and lean mass are shown. Glucose levels at different time-points from (C) oral glucose tolerance tests (OGTT) and (D) Insulin tolerance tests (ITT).(E) Oxygen consumption ($VO_2$, left panel) and Carbon dioxide generation ($VCO_2$, right panel)
*Figure 3 continued on next page*

*Figure 3 continued*

data measured by metabolic cages. The values for both light and dark cycles are shown. (**F**) Non-esterified free fatty acid (NEFA), cholesterol and triglyceride levels in serum samples from both groups. (**G**) After an oral gavage of 20% Intralipid, the clearance of serum triglycerides is shown over time for a period of 8 hr. (**H**) Lipoprotein lipase (LPL) activity measured through the LPL activity assay kit in the plasma, sWAT and eWAT from both groups. (**I**) Angiopoietin-like protein family members *Angptl3, Angptl4, Angptl8* mRNA levels in the harvested sWAT from 5-week HFD/Dox 600 mg/kg fed Adipo-rtTA and Adipo-ANG2 mice. (**J**) Representative immunoblot image (upper panel) and quantitative analysis (lower panel) of adiponectin expression in the plasma of both groups. For the western blot image, n = 3 mice per group. For all the other statistical graphs: n = 10 mice per group. Data are shown as mean ± s.e.m. *p<0.05, **p<0.01, ***p<0.001.

The following figure supplements are available for figure 3:

**Figure supplement 1.** Insulin signaling in the ANG-2 overexpressing mice.
**Figure supplement 2.** Prevention effect of ANG-2 induction against high-fat diet induced obesity.

lipase activity (*Santulli, 2014*). We observe that in sWAT of ANG-2 transgenic mice, mRNA levels of *Angptl3* and *Angptl8* are significantly reduced, while *Angptl4* merely shows a trend toward reduction (*Figure 3I*). Of note, this improved lipid metabolism results in a dramatic reduction in liver steatosis, with much less lipid accumulating in the livers of Adipo-ANG2 mice (*Figure 4B*).

To further underline the beneficial metabolic effects of enhanced ANG-2 levels, we also examined circulating adiponectin levels by Western blotting of plasma. Compared to Adipo-rtTA mice, the plasma adiponectin levels are significantly up-regulated in Adipo-ANG2 mice (*Figure 3J*). The up-regulation of circulating adiponectin reflects the overall metabolic improvements due to ANG-2 overexpression upon a HFD challenge.

In addition to the above cohort, we also examined the preventive role of ANG-2 against diet induced obesity to further provide a potential therapeutic insight for ANG-2 induction. Both control and ANG-2 transgenic mice were pre-induced by Dox chow diet for 5 weeks, and then the food was subsequently replaced by HFD (lacking Dox). In *Figure 3—figure supplement 2*, we demonstrate that the pre-induction of ANG-2 prior to HFD exposure prevents the mice from gaining body weight upon HFD exposure, and it also prevents glucose intolerance. This is particularly evident on day 3, when we observe the most dramatic improvements in glucose tolerance. By day 7, the beneficial effects start to disappear, suggesting that ongoing higher level expression of ANG-2 is required. Taken together, we conclude that either overexpressing ANG-2 during HFD challenge or pre-inducing ANG-2 prior to HFD insults can potently prevent diet-induced metabolic deficiencies.

## ANG-2 overexpression results in anti-inflammatory changes in the AT

Obesity is frequently associated with inflammation, and chronic unresolved AT inflammation is observed under obese conditions and tightly connected with AT angiogenesis (*DiSpirito and Mathis, 2015*; *Lee, 2005*). Thus, we further tested the impact of ANG-2 overexpression on changes in the AT inflammatory response. Overall, H&E staining of sWAT suggests healthier AT expansion in Adipo-ANG2 mice, with significantly smaller average adipocyte size (*Figure 4A*). Compared to control mice, macrophage infiltration, as judged by an anti-F4/80 immunostaining is slightly decreased in Adipo-ANG2 mice (*Figure 4C*), although statistical analysis did not reveal major differences (data not shown). However, a dramatic change in macrophage polarization is apparent (*Figure 4D and E*). Although general inflammatory and macrophage markers, such as *Adgre1* (gene symbol for F4/80), *Il6* and *Csf1r* (gene name for CD115), remain relatively unchanged, pro-inflammatory macrophage markers including *Ly6c1*, *Tnfa* and *Nos2* are substantially decreased; in contrast, the anti-inflammatory macrophage marker *Clec10a* (gene name for CD301) is significantly upregulated. Consistent with observations in sWAT, we show that ANG-2 overexpression significantly reduces the number of crown like structures and also induces an increase in anti-inflammatory macrophage markers in eWAT (*Figure 4—figure supplement 1*). Furthermore, flow cytometry analysis validates that there is a dramatic increase in the percentage of anti-inflammatory macrophages in both sWAT and eWAT (*Figure 4F* and *Figure 4—figure supplement 2*). This transition from a pro-inflammatory to an anti-inflammatory state of infiltrating macrophages in AT may contribute to the beneficial effects of ANG-2 toward maintaining AT healthy upon HFD insults. We also noticed that the AT shows a

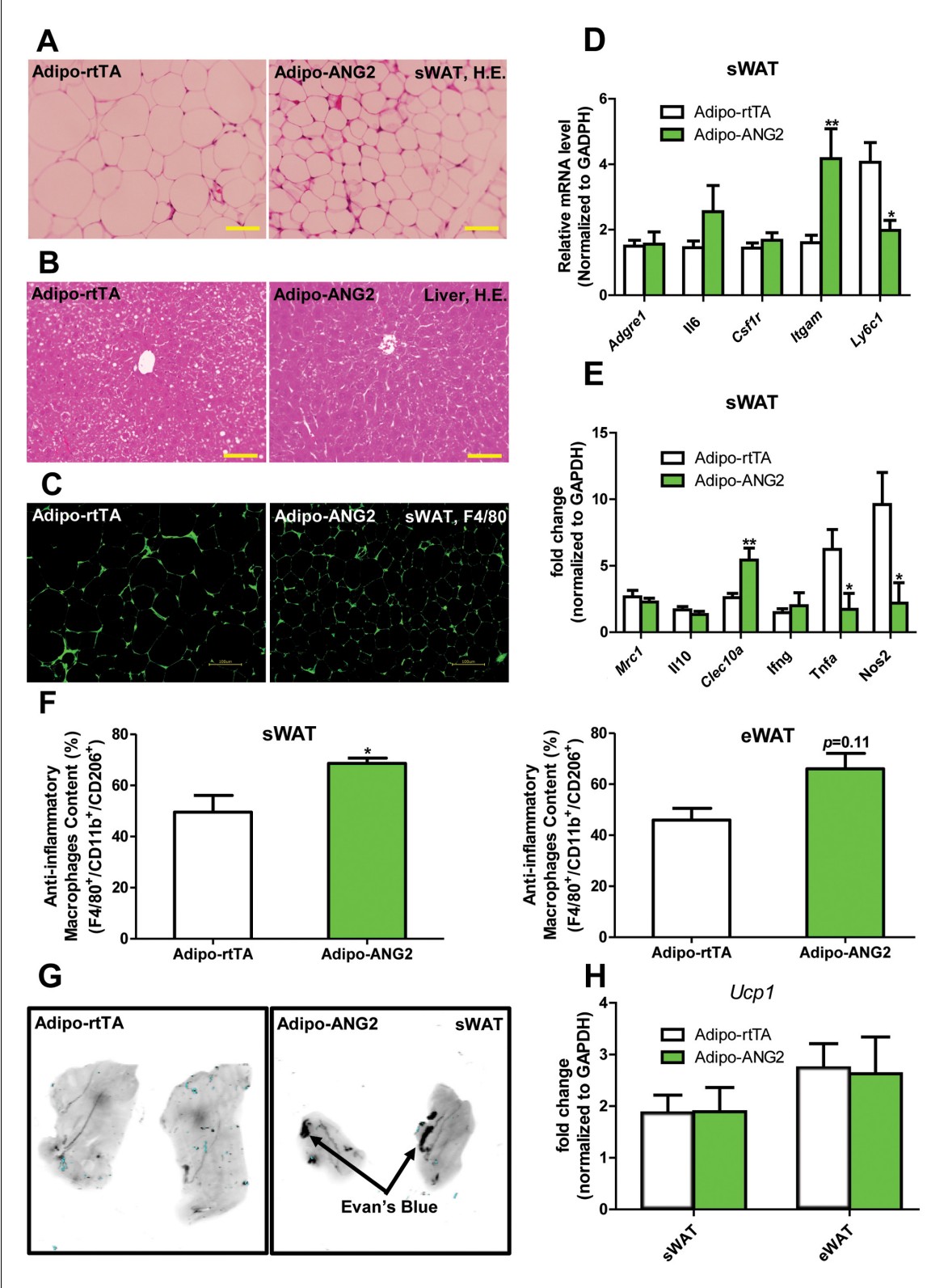

**Figure 4.** Inflammatory and histological changes after overexpression of ANG-2. (**A**) Representative H&E staining of sWAT sections (Magnification = 200×) from 5 week HFD/Dox 600 mg/kg fed Adipo-rtTA and Adipo-ANG2 mice. Bar = 60 μm. (**B**) Representative H&E staining of liver tissues (magnification = 200×) from both groups. Lipid droplets are evident in the liver sections. Bar = 60 μm.(**C**) Representative immunofluorescence staining picture of the macrophage marker F4/80 (Green, Magnification = 200×) in sWAT sections from both groups. Bar = 100 μm. (**D** and **E**). Inflammatory

*Figure 4 continued on next page*

*Figure 4 continued*

marker mRNA levels assessed by qPCR in sWAT of control and ANG-2 transgenic mice. General inflammatory markers: *Adgre1* (F4/80), *Il6* (IL-6), *Csf1r* (CD115), *Itgam* (CD11b); pro-inflammatory macrophage markers: *Ly6c, Ifng, Tnfa, Nos2*; anti-inflammatory macrophage markers: *Mrc1* (gene symbol for CD206), *Il10, Clec10a* (gene symbol for CD301). (F) The relative content of anti-inflammatory macrophages (M2 type macrophages) in both sWAT (left panel) and eWAT (right panel) from control and ANG-2 overexpressing mice. For the flow cytometry analysis, anti-inflammatory macrophages are considered as the F4/80+/CD11b+/CD206+ population, and the relative content in the total macrophage population is shown. (G) Representative Evan's Blue staining of sWAT from Adipo-rtTA and Adipo-ANG2 mice. Black arrow indicates positive staining of Evan's Blue dye, reflecting enhanced leakage from blood vessels in sWAT. (H) Beiging adipose tissue marker *Ucp1* analyzed through qPCR in sWAT and eWAT of control and ANG-2 transgenic mice. For the flow cytometry study, n = 3 mice per group. For all the other statistical graphs: n = 10 mice per group. Data are shown as mean ± s.e.m. *p<0.05, **p<0.01.

The following figure supplements are available for figure 4:

**Figure supplement 1.** Inflammation in epididymal WAT (eWAT) from ANG-2 overexpressing mice.

**Figure supplement 2.** Representative histogram of flow cytometry analysis for anti-inflammatory (M2) macrophages in ANG-2 transgenic mice.

**Figure supplement 3.** Fibrotic changes after ANG-2 overexpression.

**Figure supplement 4.** The maintenance of brown adipose tissue (BAT) in the ANG-2 overexpressing mice.

decrease in collagen accumulation, as demonstrated by trichrome staining, and a significant down-regulation in fibrotic gene expression (*Figure 4—figure supplement 3*), suggesting that along with the anti-inflammatory changes, a potent anti-fibrotic phenotype is also apparent upon ANG-2 overexpression.

Our previous studies have shown that local overexpression of VEGF-A in WAT results in improved vascularization. However, higher level expression of VEGF-A causes severely increased vascular permeability (*Sun et al., 2012*). To assess the vascular leakage in the present ANG-2 overexpression model, we injected both control mice and ANG-2 transgenic mice with Evan's Blue dye to visualize the change of vascular permeability (*Asterholm et al., 2012*; *Beard et al., 2016*). In contrast to the effects of VEGF-A, ANG-2 induction only marginally increases Evan's Blue leakage in sWAT (*Figure 4G*). This suggests that increased vascular permeability is not a significant issue in ANG-2 overexpressing mice. To further distinguish the VEGF-A and ANG-2 overexpression models, we also examined the degree of beiging in sWAT, as we and others observed widespread beiging upon VEGF-A overexpression, which prompts metabolic benefits during a HFD challenge, at least partially through the enhanced beiging program in sWAT (*Sun et al., 2012*; *During et al., 2015*). Neither sWAT nor eWAT in Adipo-ANG2 mice show elevated levels of the beiging marker uncoupling protein 1 (*Ucp1*) compared to control mice (*Figure 4H*). This suggests that the improved metabolic profile is unlikely to be the result of enhanced beiging in the ANG-2 mice. However, upon careful analysis of the BAT in the ANG-2 transgenic mice, we found that both *Vegfa* and *Ucp1* gene expressions are elevated (*Figure 4—figure supplement 4A*). Based on H&E staining of BAT sections (*Figure 4—figure supplement 4B*), control mice show a bigger lipid droplet accumulation and a more pronounced 'whitening' phenotype, while ANG-2 overexpressing mice maintain classical BAT histological features. This suggests that under HFD challenge, BAT functionality is enhanced by ANG-2 induction.

## Anti-ANG2 antibody neutralization reduces AT vascular density

The results above obtained from our Adipo-ANG2 gain-of-function model suggests a beneficial role of ANG-2 in AT expansion and function. To further validate those findings, we moved forward with a loss-of-function model. To that end, we utilized neutralizing antibodies to block endogenous ANG-2 action to examine the impact of systemic ANG-2 neutralization on HFD challenged wild-type mice, with particular focus on the influence of ANG-2 loss on AT expansion. After 5-week ANG-2 antibody injections (4 μg/g body weight, twice/week, anti-ANG2) along with HFD feeding, endothelial cell markers *Pecam1* and *Tek* in sWAT were quantitated by qPCR. Both markers are shown to be significantly down-regulated in the anti-ANG2 antibody-treated mice, when compared with control IgG administrated mice (*Figure 5A*). In addition, immunostaining with antibodies against the blood

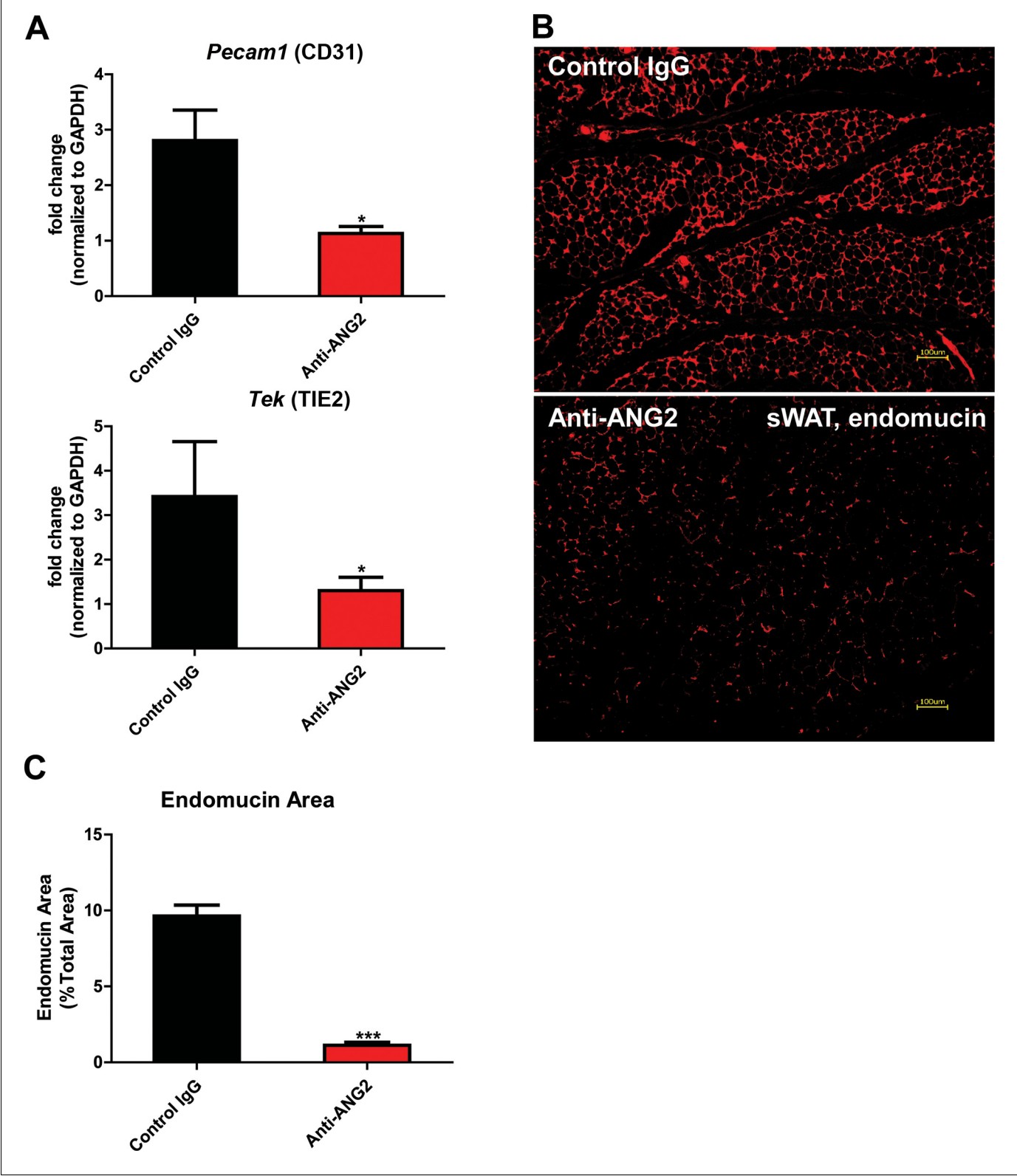

**Figure 5.** Endothelial markers and vascular staining in ANG-2 antibodies administrated mice. (**A**) Endothelial cell markers *Pecam1* and *Tek* mRNA changes in the collected sWAT from 5 week HFD stimulated wild-type mice, with administration of either control IgG or Anti-ANG2 antibody twice/ week at the dosage of 4 µg/g body weight. (**B**) Representative immunofluorescence staining image of endothelial marker endomucin (red, *Figure 5 continued on next page*

*Figure 5 continued*

magnification = 100×) and (C) the statistical result of relative endomucin area in sWAT sections from both groups. Bar = 100 μm. For all the statistical graphs: n = 5 ~ 6 mice per group. Data are shown as mean ± s.e.m. *p<0.05, ***p<0.001.

vessel marker endomucin further validates that ANG-2 neutralization prompts dramatically impaired vascular density as judged by reduced endomucin-positive areas (*Figure 5B and C*). Taken together, upon a HFD challenge, ANG-2 neutralizing antibodies cause reduced AT blood vessel formation.

## Neutralization of ANG-2 drives pro-inflammatory and pro-fibrotic changes leading to a pathological AT expansion

To further probe the relationship between reduced AT angiogenesis mediated by ANG-2 neutralization and AT expansion upon a HFD challenge, we tested the hallmarks of pathological AT expansion, including inflammation and fibrosis. We noticed that in sWAT of anti-ANG2 antibody-treated mice, the adipocyte size is slightly larger as judged by H&E staining (*Figure 6A*). In addition, significantly higher levels of fibrosis are observed as judged by trichrome staining (*Figure 6B*). Further assessment by qPCR shows that in anti-ANG2 antibody injected mice, general inflammatory markers, including *Il6* and *Adgre1*, are elevated. The upregulation can be observed for fibrotic mediators as well, such as *Tgfb1*, *Col1a1* and *Col6a1* (*Figure 6D*). Importantly, a transition from anti-inflammatory to pro-inflammatory macrophage marker expression is seen in sWAT. All pro-inflammatory macrophage markers tested, including *Ifng*, *Tnfa* and *Nos2*, are significantly induced upon ANG-2 neutralization (*Figure 6E*). These pro-inflammatory and pro-fibrotic changes represent key signatures of AT dysfunction, suggesting that blocking ANG-2 action accelerates unhealthy AT expansion upon HFD insults. Notably, the pro-inflammatory and pro-fibrotic responses were not only observed in sWAT, but also were clearly shown in eWAT of the group exposed to neutralizing antibodies (*Figure 6—figure supplement 1A–C*). In addition, we also utilized flow cytometry analysis to demonstrate that in both sWAT and eWAT, anti-ANG2 antibody neutralization causes a decrease in the content of the anti-inflammatory macrophage population (*Figure 6F* and *Figure 6—figure supplement 2*).

## Antagonizing ANG-2 exacerbates HFD induced metabolic defects

In light of the changes observed in pathological expansion of AT with the inactivation of ANG-2, we assessed the impact of ANG-2 neutralization on HFD induced metabolic changes. After 5 weeks of a HFD challenge, the body weights show no difference between the anti-ANG2 treatment group and the control IgG-treated group (*Figure 7A*). However, both glucose tolerance (*Figure 7B*) and insulin sensitivity (*Figure 7C*) are mildly, however, significantly diminished by ANG-2 antagonization, as judged by OGTTs and ITTs, respectively. The slightly decreased Akt phosphorylation after insulin injection in AT from anti-ANG2 antibody-treated mice (*Figure 7—figure supplement 1*) further suggests that ANG-2 neutralization impairs insulin signaling and promotes insulin resistance. Furthermore, although the serum lipid profile remains unchanged between antibody and control groups (data not shown), an oral lipid gavage study demonstrates significantly impaired lipid clearance in ANG-2 antibody-treated mice (*Figure 7D*). The impaired lipid disposal is associated with a massive elevation of triglyceride levels in the liver (*Figure 7E*). This is further validated by H and E staining of liver slides (*Figure 6C*), highlighting higher lipid droplet accumulation in the liver of antibody-treated mice. In addition, the endogenous LPL inhibiting genes *Angptl3, Angptl4* and *Angptl8* are all elevated after anti-ANG2 antibody treatment (*Figure 7F*), suggesting a likely mechanism for a decrease in LPL activity, thereby leading to impaired lipid clearance upon Ang-2 inactivation. Lastly, immunoblotting for plasma adiponectin (*Figure 7G*) demonstrates that ANG-2 neutralization significantly lowers circulating adiponectin levels, reflecting the systemic metabolic deterioration seen in the ANG-2 neutralizing mice.

## Discussion

AT is a central pillar for the maintenance of metabolic homeostasis. This is not only due to its ability to store lipids and to secrete numerous adipokines, but also due to its extraordinary plasticity while

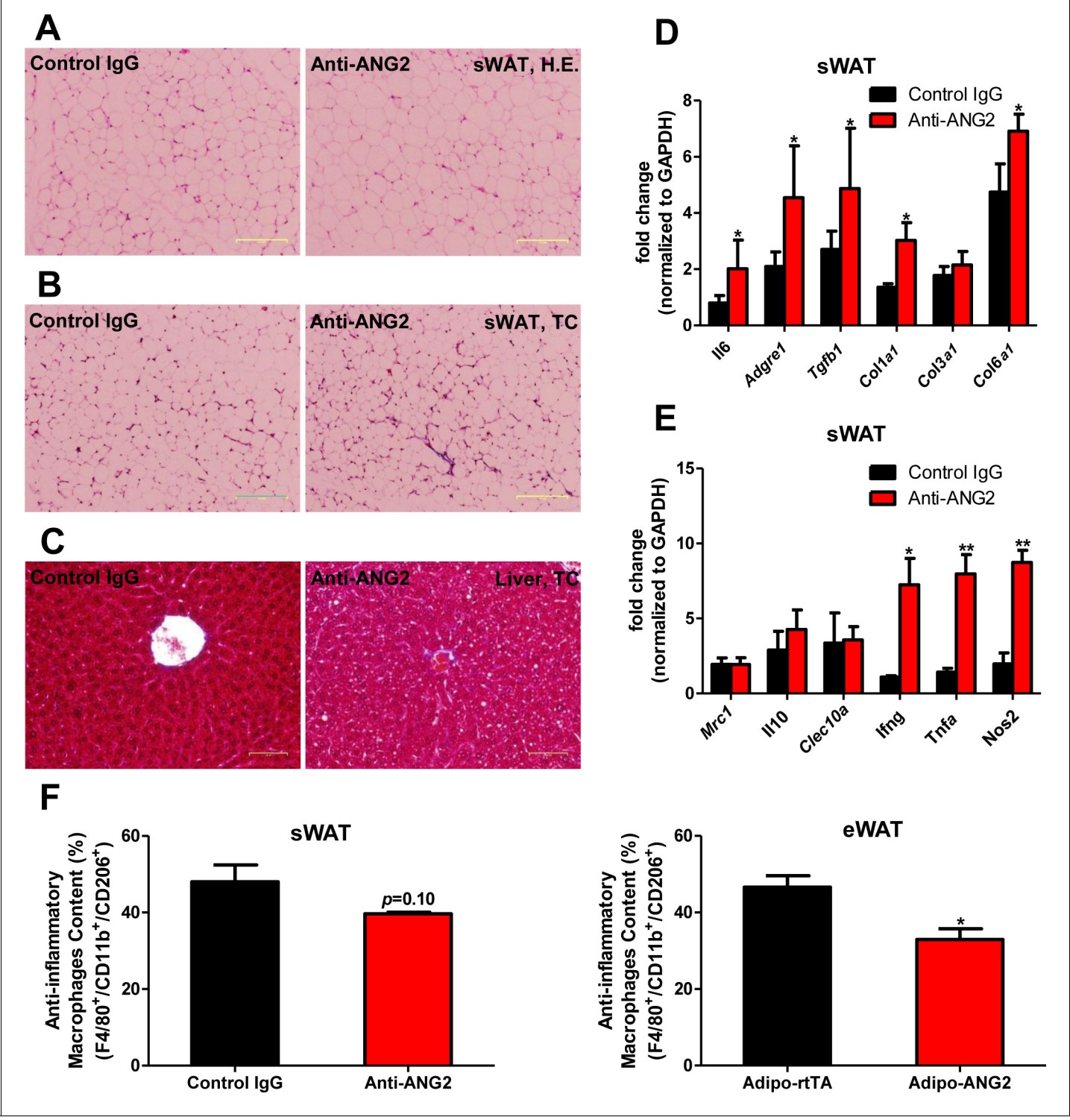

**Figure 6.** Inflammatory and fibrotic changes as well as histological features after ANG-2 neutralization. (A) Representative H.E. staining image of sWAT sections (Magnification = 100×) from 5 week HFD fed mice with control IgG or Anti-ANG2 antibody injection. Bar = 160 μm. (B) Representative Trichrome (TC) staining pictures of sWAT sections (Magnification = 100×) from both groups. Fibrotic changes are demonstrated as the blue staining. Bar = 160 μm. (C) Representative TC staining image of liver tissues (Magnification = 200×) from both groups. Lipid droplets are shown the liver sections. Bar = 60 μm. (D and E) Transcriptional levels of key inflammatory markers and crucial pro-fibrotic molecules measured through qPCR in sWAT from control IgG and Anti-ANG2 mice. General inflammatory markers: *Adgre1, Il6, Csf1r, Itgam*; pro-inflammatory macrophage markers: *Ly6c, Ifng, Tnfa, Nos2*; anti-inflammatory macrophage markers: *Mrc1, Il10, Clec10a*; fibrotic molecules: TGF-β1, Collagen 1α1 (*Col1a1*), *Col3a1, Col6a1*. (F) The

*Figure 6 continued on next page*

*Figure 6 continued*

relative content of anti-inflammatory macrophages (M2 type macrophages) in both sWAT (left panel) and ANG-2 antibody-injected mice. Through the flow cytometry analysis, anti-inflammatory macrophages are considered as the F4/80+/CD11b+/CD206+ population, and the relative percentage in the total macrophage population is shown. For the flow cytometry study, n = 3 mice per group. For all the statistical graphs: n = 5 mice per group. Data are shown as mean ± s.e.m. *p<0.05, **p<0.01.
The following figure supplements are available for figure 6:

**Figure supplement 1.** Histological and inflammatory features in eWAT after ANG-2 neutralization.
**Figure supplement 2.** Representative histogram of flow cytometry analysis for anti-inflammatory (M2) macrophages in the ANG-2 neutralizing mice.

exposed to various metabolic challenges. Many individuals expose themselves to nutritional and caloric excess. Upon excessive energy intake, AT undergoes rapid expansion, initially as a protective response to spare other tissues excess lipotoxic exposure. Chronically, this challenges AT and causes pathological changes, leading to systemically unfavorable metabolic changes. Considering that the rapid expansion of AT requires an appropriate vascular infrastructure, the regulation and mechanics of angiogenesis become important and integral components toward understanding how AT function is properly maintained (*Sun et al., 2011*; *Cao, 2013*). A number of pro-angiogenic factors have previously been described, such as leptin, adiponectin, various VEGF family members, ANG-1, fibroblast growth factor-2 (FGF-2) and hepatocyte growth factor (HGF). These AT-derived factors participate in the angiogenesis process and exert important metabolic effects (*Sun et al., 2011*; *Cao, 2013*; *Sung et al., 2013*; *Elias et al., 2012*). However, the relative importance of ANG-2 in the context of AT angiogenesis has not yet been addressed. In the present study, we use an adipocyte-specific, Dox-inducible ANG-2 overexpression mouse model to directly test the relevance of ANG-2 in vascularization of AT and whether ANG-2 exerts beneficial metabolic effects against diet-induced obesity. Our results demonstrate that ANG-2 overexpression induces a pro-angiogenic program in WAT, thereby protecting against HFD-induced metabolic challenges. Our loss-of-function antibody neutralization studies further confirm the beneficial effects of endogenous ANG-2 toward maintenance of sufficient AT vasculature and protection from HFD-induced metabolic defects. Other reports implicate leptin, considered to be a pro-angiogenic factor secreted by AT, as a potent inducer of ANG-2 expression (*Cohen et al., 2001*); in addition, Xue and colleagues demonstrate that FOXC2 overexpression increases vessel sprouting in WAT, potentially through an upregulation of ANG-2 (*Xue et al., 2008*). Consistent with the above findings, we directly demonstrate here that ANG-2 exerts a pro-angiogenic role during HFD induced AT expansion, leading to improvements in adipose tissue physiology.

We not only demonstrate that ANG-2 mediates angiogenesis in WAT, but also conclude that the enhanced angiogenic process triggers improvements in metabolic effects upon HFD insults. Mechanistically, increasing vascular function and decreasing AT inflammation contribute to the beneficial effects of ANG-2. Angiogenesis and inflammation are not only closely associated, but also mutually regulated processes in AT. Furthermore, ANG-2 is considered to be an important mediator participating in both processes, not only modulating angiogenesis, but also regulating inflammatory responses (*Scholz et al., 2015*). However, it is still debatable whether ANG-2 exerts actual pro-inflammatory effects or rather plays an anti-inflammatory role. Several studies suggest that ANG-2 can recruit innate immune cells through both Tie2-dependent and Tie2-independent processes (*Fiedler et al., 2006*). In contrast, other studies demonstrate that in mice overexpressing ANG-2, tissues show increased myeloid cell infiltration without active tissue damage (*Scholz et al., 2011*). In addition, ANG-2 prompts macrophages to produce high levels of the anti-inflammatory cytokine IL-10 (*Coffelt et al., 2011*); ANG-2 can promote macrophage conversion into an anti-inflammatory phenotype, as defined by the induction of IL-10, CD206 and a decrease of IL-6 and TNF-α (*Seok et al., 2013*; *Kurniati et al., 2013*). Our current study supports and extends the concept that ANG-2 is an anti-inflammatory factor in AT expansion, since we identify that AT-based overexpression of ANG-2 diminishes pro-inflammatory markers and elevates anti-inflammatory markers and leads to additional beneficial effects during healthy AT expansion.

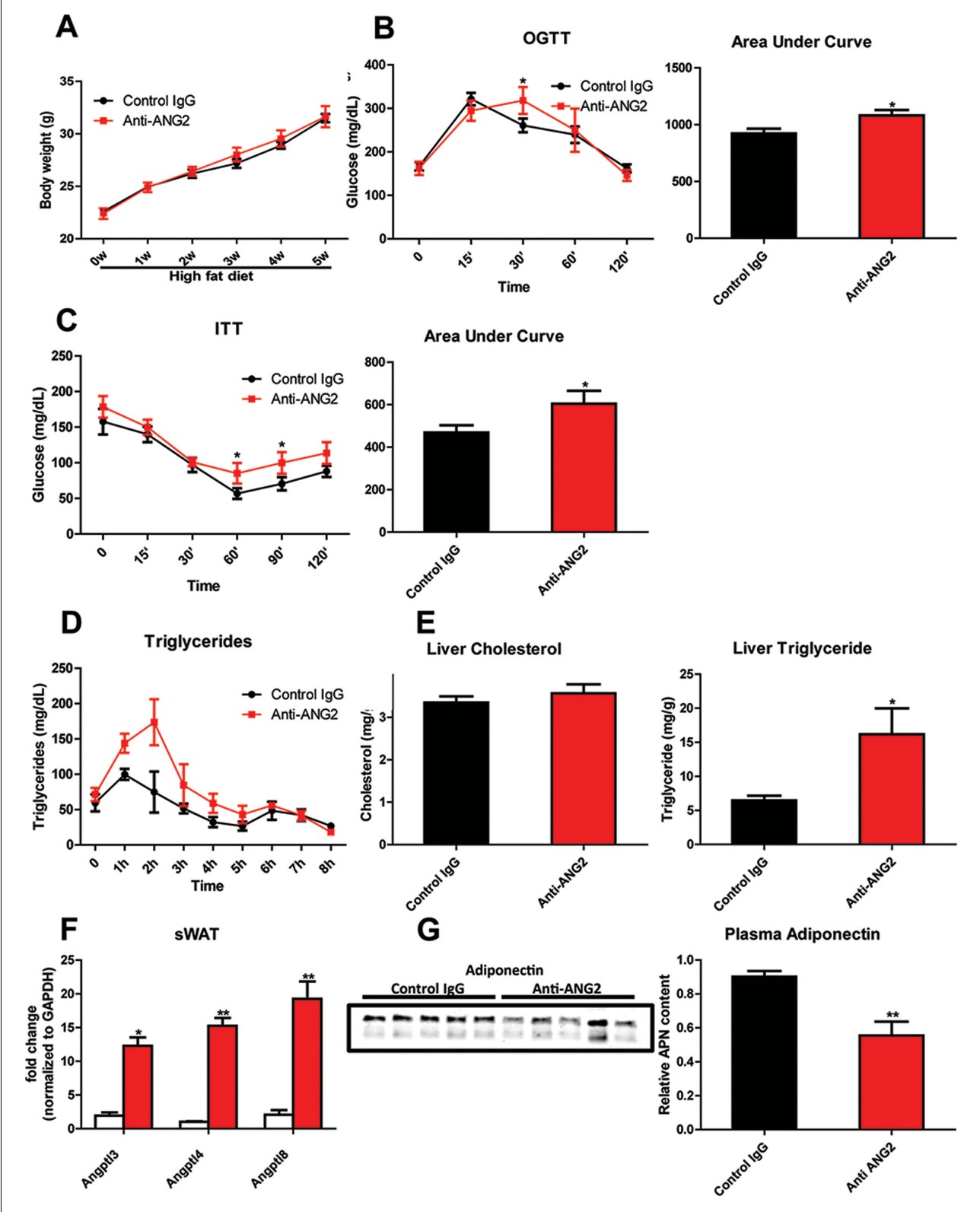

**Figure 7.** Systemic metabolic effects after ANG-2 antibody injections. During HFD challenges for five weeks in the wild-type mice, which were administrated with either 4 μg/g body weight control IgG or anti-ANG2 antibody and the mice then underwent the following metabolic analyses: (A) Body weight was monitored for five weeks.(B) Glucose levels over different time-points during an OGTT (left panel) and quantification of areas under curve (right panel) presented as the mean area under each glucose level curve. (C) The insulin sensitivity demonstrated by ITT (left panel) and

*Figure 7 continued on next page*

*Figure 7 continued*

quantification of area under the curve (right panel) from both groups.(D) After an oral gavage of 20% Intralipid, the clearance of serum triglycerides is shown over time for a period of 8 hr. (E) Liver cholesterol (left panel) and triglycerides levels (right panel) from both groups. (F) Angiopoietin-like protein family members *Angptl3, Angptl4 and Angptl8* mRNA levels in sWAT from 5-week HFD fed control IgG and Anti-ANG2-treated mice.(G) Representative immunoblot images (left panel) and quantitation (right panel) of adiponectin expression in the plasma of both groups. For all the statistical graphs: n = 5 mice per group. Data are shown as mean ± s.e.m. *p<0.05, **p<0.01.
The following figure supplement is available for figure 7:

**Figure supplement 1.** Insulin signaling in sWAT after ANG-2 antibody administration.

ANG-2 and its receptor Tie-2 have been intensively studied in the tumor angiogenesis field. However, the detailed consequences of ANG-2 signaling mechanisms remain challenging to pinpoint. The complexity of the ANG-2/Tie-2 ligand/receptor system is reflected by the difficulty to determine whether ANG-2 is an agonist or antagonist of Tie-2. This is very much context-dependent (*Felcht et al., 2012*). Under different conditions, ANG-2 can either enhance Tie-2 signaling to promote vessel remodeling, or it can suppress phosphorylation of Tie-2 and its downstream targets to cause vessel instability and regression. Two recent reports provided insightful explanations for the ANG-2/Tie-2 signaling dilemma (*Kim et al., 2016*; *Korhonen et al., 2016*). One study suggested that under normal conditions, ANG-2 acts as a Tie-2 agonist to induce angiogenesis. However, during infections, ANG-2 exerts antagonistic effects on Tie-2 due to a reduction of Tie-1 expression (*Kim et al., 2016*). Independently, another report indicates that interactions between Tie-1 and Tie-2 play an important role in the activation of Tie-2 by ANG-2. When the interaction domain of Tie-1 with Tie-2 is cleaved off during acute endotoxemia, the ANG-2 agonist activity is blunted (*Korhonen et al., 2016*). Our present study fully supports the pro-angiogenic effects of ANG-2 in AT. We observe an elevation of Tie-2 upon ANG-2 overexpression, which is surprising and looks like a feed-forward type of response.

The beneficial effects of ANG-2 mediated angiogenesis on the protection against HFD induced metabolic defects, as demonstrated by our present study, may outline additional therapeutic avenues for type 2 diabetes. In humans, a meta-analysis demonstrated an association between angiogenesis and insulin sensitivity, showing that a chronic insufficiency in tissue vascularization contributes to insulin resistance (*Shungin et al., 2015*). Therefore, activating endogenous ANG-2 or the delivery of recombinant ANG-2 could be an effective approach, though to do this in an adipose tissue-specific way is obviously challenging. Similarly, we have previously shown that VEGF-A overexpression significantly increases WAT vascularization and 'beiging', and the associated enhancement of energy expenditure prevents the unfavorable metabolic changes upon a HFD challenge (*Sun et al., 2012*, *2014*). A recent study also showed that a peptide-functionalized nanoparticle delivering rosiglitazone or prostaglandin E2 (PGE2) to the AT vasculature promotes angiogenesis and causes significant browning of WAT, triggering further reduced weight gain and improvements in insulin sensitivity in a diet-induced obese mouse model (*Xue et al., 2016*). In contrast to the VEGF-A overexpression model, ANG-2 induction does not activate a beiging program in WAT, but confers improvements in metabolic homeostasis in a beiging-independent manner. Considering the fact that ANG-2 induces WAT angiogenesis without beiging, we conclude that enhancement of vasculature per se is necessary, but not sufficient to facilitate development of beiging. This reflects one of the most important differences between the VEGF-A overexpression model and the ANG-2 overexpression model in AT. Furthermore, at higher concentrations, enhancing VEGF-A also triggers widespread edema formation, a phenomenon we did not observe with ANG-2 overexpression, although a slight increase in vascular permeability was shown.

However, whether pro-angiogenic or anti-angiogenic therapies in obesity should be employed is still a matter of debate. Because angiogenesis facilitates AT expansion, anti-angiogenic approaches were originally proposed to be effective anti-obesity approaches. Angiostatin, endostatin and thalidomide, which are all angiogenesis inhibitors, significantly reduce body weight in genetically and diet-induced obese mouse models (*Rupnick et al., 2002*). TNP-470, a blocking antibody targeting VEGFR2, also has anti-obesity effects in both diet-induced and *ob/ob* mice (*Bråkenhielm et al., 2004*). Furthermore, several compounds and natural products with anti-angiogenic properties exert

anti-obesity and insulin sensitizing effects (*Luo et al., 2016*; *Klaus et al., 2005*; *Park et al., 2015*). Counterintuitively, both promoting and suppressing angiogenesis can protect against obesity and associated insulin resistance; the paradoxical outcomes depend on the stage of AT expansion and the extent of obesity. Intervening at late stages of obesity, 'choking off' large adipocytes that tend to be more inflamed with anti-angiogenic approaches seems to be the major source of the improvements. In our present study, in the early stages of obesity (5-week HFD challenge), enhancing angiogenesis by ANG-2 induction is clearly an effective therapy against diet-induced weight gain and insulin resistance (*Figure 8*).

## Materials and methods

### Animals

We generated a mouse model with Dox inducible, adipocyte-specific ANG-2 overexpression by crossing the TRE-ANG2 transgenic mice with adiponectin promoter driven-rtTA transgenic mice (Adipo-rtTA). In the ANG-2 antibody neutralization study, C57BL/6 wild-type mice (RRID: MGI: 3038857) were utilized. All experiments were conducted using littermate-controlled male mice and were started at the age of 8 weeks. All experimental mice were housed in a barrier animal facility with 12 hr dark-light cycle with free access to water and food. During HFD challenge experiments together with Dox induction, mice were fed with a paste diet containing 60% caloric from fat supplemented with 600 mg/kg Dox (S7067; Bio-Serv). In the antibody neutralization experiments, regular HFD with 60% caloric from fat (D12492, Research Diets) were utilized. All animal experiments conducted in the present study were approved by the Institutional Animal Care and Research Advisory Committee at the University of Texas Southwestern Medical Center (APN# 2015–101207).

### Reagents

For ANG-2 antibody neutralization experiments, antibody was supplied by AdipoGen, which is a mouse IgG2bλ (anti-Angiopoietin-2, mAb (rec.) (blocking) (Angy-2–1) (AG-27B-0016PF, AdipoGen Life Sciences, RRID: AB_2490509). Anti ANG-2 antibody (ab8452, RRID: AB_306569) for Western blotting and immunofluorescence staining was purchased from Abcam (Cambridge, MA). An antibody against $\beta$-actin (A3854, RRID: AB_262011) was from Sigma-Aldrich (St. Louis, MO). Anti-endomucin (sc-65495, RRID: AB_2100037) and anti F4/80 (sc-25830, RRID: AB_2246477) primary

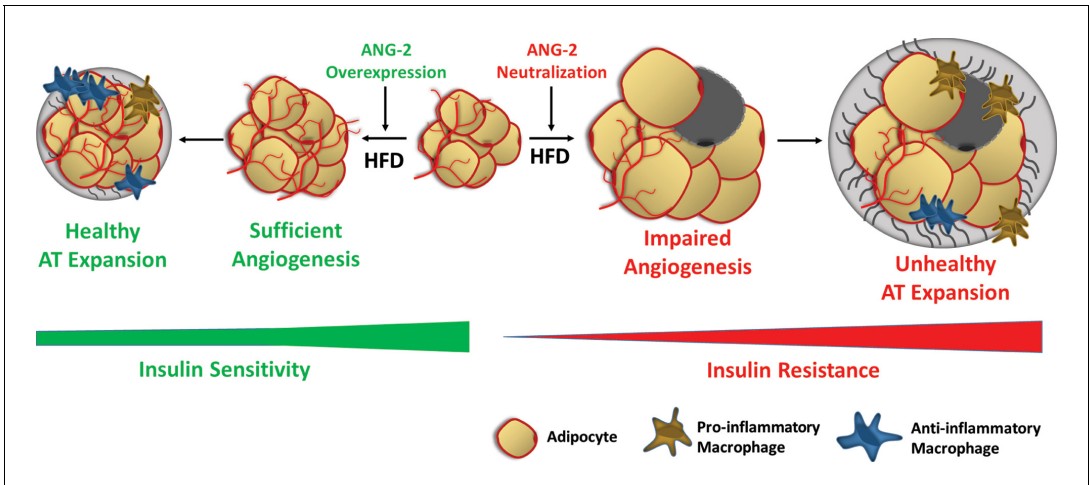

**Figure 8.** Schematic illustration of the role of ANG-2 during AT expansion and modulation of insulin sensitivity. When ANG-2 is genetically overexpressed in the WAT upon a HFD challenge, it results in sufficient angiogenesis and leads to a further healthy AT expansion with signature improvements of vasculature and anti-inflammatory macrophage infiltration, associated with maintenance of insulin sensitivity. Conversely, during HFD feeding, if the endogenous ANG-2 is neutralized by antibodies, the AT demonstrates an unhealthy expansion pattern with features of impaired vascularization and enhanced pro-inflammatory as well as increased pro-fibrotic changes, eventually giving rise to insulin resistance.

antibodies were obtained from Santa Cruz Biotechnology, Inc. (Santa Cruz, CA). Perilipin antibody (20R-PP004) was obtained from Fitzgerald, Inc. (Acton, MA). Anti-HIF-1α primary antibody (NB100-105, RRID: AB_10001154) was purchased from Novus Biologicals (Littleton, CO). Primary antibodies against GAPDH (#2118S, RRID: AB_561053), phosphorylated Akt (pAkt, Ser473, #9271S, RRID: AB_329825) and total Akt (#2920S, RRID: AB_1147620) came from Cell Signaling Technology, Inc. (Beverly, MA). Unless specifically indicated, all other reagents were obtained from Sigma-Aldrich Corporation (St. Louis, MO).

## RNA extraction and real-time quantitative PCR (qPCR)

Different depots of adipose tissues, including subcutaneous WAT, epididymal WAT and brown AT, as well as heart, kidney and liver tissues were harvested and quickly frozen in liquid nitrogen for future use. For the total RNA extraction, a combination of Trizol (Invitrogen, Carlsbad, CA) reagents and the RNeasy RNA extraction kit (#74106, Qiagen, Valenica, CA) was utilized. Briefly, after homogenizing the tissues by using a TissueLyser (Qiagen), the RNAs were isolated following the protocol from the RNeasy kit. Then the quality and concentration of the RNA were determined through the Nanodrop Spectrophotometer (N1-1000, Thermo Scientific, Wilmington, DE). A total of 1 µg RNA underwent subsequently reverse transcriptional reactions with an iScript cDNA synthesis kit (#170–8891, Bio-Rad Laboratories, Inc., Hercules, CA). cDNAs were stored and utilized for further qPCR analysis of relative gene expressions. Briefly, we utilized the SYBR Green PCR Master Mix (#4309155, Life Technologies, Carlsbad, CA) to perform the qPCR reactions, and the experiments were conducted on an ABI Prism 7900HT Real-Time PCR System (Applied Biosystems, Foster City, CA). All the primer sequences in this study were validated in previous studies and are listed in *Supplementary file 1*, and *Gapdh* was used as the internal control.

## Body composition measurements

The precise measurements of mouse whole body compositions including total body fat and lean mass were performed through the Bruker Minispec mq10 system (Bruker Corporation, Billerica, MA).

## Systemic metabolic tests

Systemic metabolic tests in this study included OGTT, ITT and lipid clearance study. For the OGTTs, 5-hr fasting was conducted prior to administration of 2.5 g/kg body weight glucose to the mice through oral gavage. Blood was collected from tail veins at each time point indicated, and then the blood samples were centrifuged at 6000 rpm for 15 min and stored for further measurements. Glucose concentrations were measured by using an oxidase-peroxidase-based colorimetric assay (Sigma-Aldrich, St. Louis, MO). For the ITTs, mice underwent 5 hr fasting before 1 µ/kg body weight of insulin was administered through intraperitoneal (i.p.) injection. Blood samples were collected at the indicated time points and utilized for glucose measurement. The lipid clearance test was performed by oral gavage of 20% Intralipid (NOC0338-0519-58, Baxter Healthcare Corporation, Deerfield, IL) to the mice after an overnight fast. Blood samples were collected every hour for a continuous 8-hr period, the serum samples obtained were utilized for triglyceride measurement through a colorimetric kit (Infinity, Thermo Scientific, Wilmington, DE).

## Metabolic cage studies

The metabolic cage studies were performed by The University of Texas Southwestern Medical Center Metabolic Phenotyping Core facility. Briefly, the mice were maintained on a 12 hr dark/light cycle at room temperature. Metabolic parameters including oxygen consumption and carbon dioxide generation were monitored and obtained continuously using the TSA calorimetric system (TSA System, Germany) as described before (*Sun et al., 2014*). During the whole process, all ANG-2 transgenic mice and their littermate controls were single-housed in the metabolic chambers and kept on HFD/Dox 600 mg/kg and water ad libitum.

## Serum and liver chemistry

When sacrificing the mice, whole blood samples were obtained and centrifuged at 6000 rpm for 15 min to collect the serum. Samples were analyzed for cholesterol, triglycerides and free fatty acid

(NEFA) measurements through a Vitros 250 Chemistry system (Ortho Clinical Diagnostics, Raritan, NJ).

Similarly, after ~150 mg of frozen liver tissues were homogenized, the lysates were utilized for tissue cholesterol and triglycerides measurements after extraction.

## Isolation of stromal vascular fraction (SVF) and flow cytometry analysis

Subcutaneous and epididymal adipose tissues were obtained, rinsed with PBS, and then completely minced. Tissues were digested for one hour at 37°C with standard digestion buffer (100 mM Hepes PH 7.4, 120 mM NaCl, 50 mM KCl, 5 mM glucose, 1 mM CaCl$_2$, 1.5% bovine serum albumin, and 1 mg ml$^{-1}$ collagenase D (Roche, Mannheim, Germany). After digestion, the mixture was passed through 100 µm cell strainers and then centrifuged at 600 g for five minutes. After discarding the floating adipocytes, the pelleted stromal vascular cells were re-suspended in PBS with 2% FBS. The cells were filtered with 40-µm cell strainers and spun at 600 g for five minutes, and then the re-suspended SVF cells were incubated with red blood cell lysis buffer (BD Biosciences, San Jose, CA) to lyse red blood cells. After washing and blocking in PBS with 2% FBS containing purified anti-mouse CD16/CD32 Fc Block (1:200) (BD Biosciences), the SVF cells were ready for antibody incubation and flow cytometry analysis.

Flow cytometry analysis was utilized to measure the ratio between anti-inflammatory (M2) and pro-inflammatory (M1) macrophages in the white adipose tissues. The isolated SVF cells were incubated with primary antibodies for 30 min at 4°C for different markers: anti-CD45 (Cat# 103106, RRID: AB_312971), anti-CD11b (Cat# 101212 RRID: AB_312795), anti-F4/80 (Cat# 123108 RRID: AB_893502), anti-CD206 (Cat# 141716 RRID: AB_2561992), and anti-CD11c (Cat# 117320 RRID: AB_528736) (Biolegend, San Diego, CA). After washing three times with PBS containing 2% FBS, cells were then analyzed using a FACSCalibur flow cytometer at UT Southwestern Medical Center Flow Cytometry Core Facility.

## Histological analysis

Mouse adipose tissue and liver paraffin-embedded sections were cut at 4 µm and stained with hematoxylin and eosin (H.E.) as well as Masson's Trichrome C staining, which were performed by the University of Texas Southwestern Medical Center Histology Core. After images (100× or 200× magnification) were recorded by the FSX100 Inverted Microscope (Olympus, Waltham, MA), histomorphological changes were analyzed.

## Immunofluorescence and immunohistochemistry staining

Paraffin-embedded slides were utilized for immunostaining. Briefly, deparaffinized sections were stained with anti-ANG2, anti-F4/80 and anti-perilipin primary antibodies and incubated overnight. Then the corresponding fluorescent labeled secondary antibodies (Life Technologies, Carlsbad, CA) were added and finally the sections were counterstained with DAPI. The pimonidazole hypoxia probe staining was conducted according to the protocol of a commercially available kit (Hypoxyprobe-1 Plus Kit, Hypoxyprobe, Inc., Burlington, MA), followed by DAB staining. The fluorescence or histochemistry pictures were acquired with an Olympus FSX100 Microscope and analyzed by the Image Pro Plus software (Media Cybernetics, Rockville, MD).

## Vascular permeability assay

Vascular permeability was assessed by Evan's Blue injections as previously reported (*Asterholm et al., 2012*; *Beard et al., 2016*). Briefly, the mice were administrated 160 mg/kg Evan's Blue dye by tail vein injection. After 30 min of Evan's Blue in circulation, the mice were subjected to anesthesia followed by perfusion with PBS. To visualize Evan's Blue in adipose tissue, whole inguinal fat pads were harvested and fixed in 4% paraformaldehyde overnight. After fixation and rinsing with PBS, the fat pad was imaged for Evan's Blue autofluorescence at 700 nm by utilizing a LI-COR Odyssey Imager (LI-COR, Lincoln, NE).

## LPL activity test

The lipoprotein lipase (LPL) activity was measured via a commercial LPL assay kit (STA-610, Cell BioLabs, Inc., San Diego) as described previously (*Kusminski et al., 2012*). Briefly, after homogenizing frozen mouse adipose tissues in a buffer containing protease inhibitors and centrifugation, the tissue

lysates and serums were used for LPL activity assays. After incubation with reagents in the assay kit according to the manufacturer's instructions, the fluorescence signal (excitation in 480–485 nm and emission in the 515–525 nm) was read and recorded with a fluorescence microplate reader (POLAR-star OPTIMA, BMG LABTECH, Ortenberg, Germany). The activity of LPL was then calculated following instructions of the kit.

## Western blot analysis

After frozen adipose tissue samples were homogenized, they were lysed in RIPA buffer (1% Triton-X100, 50 mmol/L Tris-HCl (pH 8.0), 0.25 mol/L NaCl, 5 mmol/L EDTA) containing PMSF (phenylmethyl-sulfonyl fluoride) and protease inhibitor cocktail (Calbiochem, San Diego, CA, USA). Following centri-fugation at 12,000g for 15 min at 4°C, the supernatants were collected and protein content was measured by the bicinchoninic acid (BCA) assay kit (Thermo Scientific Pierce, Rockford, IL). Briefly, tissue samples (40 µg) or serum (0.1 µl diluted in 10 µl Tris-buffered saline Tween (TBS-T, 20 mmol/L Tris, 137 mmol/L NaCl, pH 7.6) were loaded and separated by 4–12% sodium dodecyl sulfate-polyacryl-amide gel (SDS-PAGE). Upon completion of the electrophoresis, the proteins were transferred to a PVDF membrane. Non-specific binding sites were blocked by pre-incubating the membrane in 5% skimmed milk in TBS-T. The membranes were then incubated overnight at 4°C with specific primary antibodies: anti ANG-2 (1:500), anti $\beta$-actin (1:3000), anti adiponectin (1:2000), anti HIF-1$\alpha$ (1:1000), anti GAPDH (1:2000), anti p-Akt (Ser473) (1:1000) or anti total Akt (1:1000). After washing, fluores-cent-conjugated secondary antibody (IRDye, LI-COR, Lincoln, NE) were added and incubated. Then the Western blot bands were scanned by the LI-COR Odyssey Imager (LI-COR, Lincoln, NE), and the band intensity was analyzed through the LI-COR Odyssey Imager software.

## Statistical analysis

All data are presented as means ± s.e.m. We utilized GraphPad Prism 5.0 (GraphPad Software, Inc., La Jolla, CA, USA) to perform the statistical analyses. For comparisons between two independent groups, a Student's t-test was used. Differences between two groups over time were determined by a two-way analysis of variance (ANOVA) followed by a Bonferroni post-test to compare replicate means in each time point. The box-and-whisker analysis was performed to exclude potential outliner data accordingly. $p < 0.05$ was considered statistically significant. All the detailed statistical methods, sample sizes and p-values are listed in the *Supplementary file 2*.

## Acknowledgements

We thank Dr. Bob Hammer and the Transgenic Core Facility at UTSW for the generation of the trans-genic lines and the Metabolic Phenotyping Core, Pathology Core, and Flow Cytometry Core at UTSW for their excellent experimental assistance. We are also grateful to Dr. Vivian Paschoal in the Touchstone Diabetes Center at UTSW for her guidance and assistance in data analysis. This study was supported by US National Institutes of Health (NIH) grants P01-DK088761, R01-DK55758 and R01-DK099110 (PES). The funding agencies had no roles in study design, data collection and inter-pretation, or the decision to submit the work for publication.

## Additional information

### Funding

| Funder | Grant reference number | Author |
| --- | --- | --- |
| National Institutes of Health | P01-DK088761 | Philipp E Scherer |
| National Institutes of Health | R01-DK55758 | Philipp E Scherer |
| National Institutes of Health | R01-DK099110 | Philipp E Scherer |

The funders had no role in study design, data collection and interpretation, or the decision to submit the work for publication.

## Author contributions
YAA, Formal analysis, Investigation, Methodology, Writing—original draft; KS, Conceptualization, Investigation, Methodology, Writing—original draft; NJ, Formal analysis, Investigation; FZ, Investigation, Methodology; YD, Resources, Investigation; OD, Resources, Writing—review and editing; CMK, Conceptualization, Resources; PES, Conceptualization, Supervision, Funding acquisition, Writing—review and editing

## Author ORCIDs
Yu A An, http://orcid.org/0000-0002-9678-3382
Yingfeng Deng, http://orcid.org/0000-0003-1314-5105
Philipp E Scherer, http://orcid.org/0000-0003-0680-3392

## Ethics
Animal experimentation: All animal experiments conducted in the present study were approved by the Institutional Animal Care and Research Advisory Committee at the University of Texas Southwestern Medical Center (APN# 2015-101207).

## Additional files

### Supplementary files
• Supplementary file 1. Primer sequences for qPCR.

• Supplementary file 2. Statistical information.

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
