## [Decision Letter]

Thank you for submitting your article "Enhanced Angiogenesis by Angiopoietin-2 in White Adipose Tissue is Causing an Improved Metabolic Phenotype" for consideration by *eLife*. Your article has been reviewed by 3 peer reviewers, and the evaluation has been overseen by Peter Tontonoz as the Reviewing Editor and Fiona Watt as the Senior Editor.

The reviewers have discussed the reviews with one another and the Reviewing Editor has drafted this decision to help you prepare a revised submission.

Summary:

This paper tests hypotheses regarding adipose tissue vascularization and expansion and their relationship to systemic metabolic homeostasis. The authors study the effects of gain or loss of ANG-2 expression in adipose tissue on glucose tolerance and energy expenditure. The data show that increased adipose tissue vascularization stimulated by ANG-2 expression is beneficial, while blockade of ANG-2 by antibody administration is detrimental.

The reviewers were in agreement that the work was potentially of interest to the general audience of *eLife*. The studies were judged to be technically sound and the reviewers believed that the majority of the conclusions drawn were supported by the data. Although the concept that adipose vascularization is important for healthy adipose expansion is not new, the paper builds on prior knowledge to move the field forward by providing additional rigorous experimental support for the concept. At the same time, the review process identified several opportunities to strengthen the work.

Essential revisions:

1) Obesity-induced metabolic complications are highly associated with inflammation in visceral fat tissues including eWAT. It would be important to examine the effects of ANG-2 overexpression on inflammation in eWAT. Also, it would be helpful to have FACS analysis of macrophage populations in adipose tissue for both the gain and loss of ANG-2 function studies.

2) The data shows that ANG2 prevents diet-induced body weight gain. It would be exciting to test the therapeutic effect of ANG2; that is to determine if the inducible expression of ANG2 in obese mice sufficiently reverse obesity-linked glucose intolerance, inflammation, and lipid profile. If such studies have already been initiated and are well along, it would be great to include them. However, the reviewers recognize that starting such an experiment at this stage is beyond the scope of an *eLife* revision.

3) Did ANG-2 overexpression reduce adipose tissue hypoxia (Figure 2)? The authors could examine HIF-1α protein and target gene expression in sWAT and eWAT. Pimonidazole staining would be helpful to access the degree of adipose tissue hypoxia upon ANG-2 overexpression.

4) It would be insightful to determine if reduced adipose inflammation by ANG2 tg expression was associated with reduced adipose fibrosis.

5) The authors may wish to comment on the possibilities as to how the ANG2tg mice exhibited higher energy expenditure though ANG2 did not induce browning. Does ANG2 expression activate BAT thermogenesis?

6) Was insulin signaling cascade improved in adipose tissue of ANG-2 tg or ANG-2 neutralized mice (Figure 3 and Figure 7) upon HFD feeding?

---

## [Author Response]

*Essential revisions:*

*1) Obesity-induced metabolic complications are highly associated with inflammation in visceral fat tissues including eWAT. It would be important to examine the effects of ANG-2 overexpression on inflammation in eWAT. Also, it would be helpful to have FACS analysis of macrophage populations in adipose tissue for both the gain and loss of ANG-2 function studies.*

We thank the reviewers for the insightful comments.Following your suggestions, we performedH&E staining and qPCR analysis of inflammatory genes from the visceral fat tissues (eWAT) in ANG-2 overexpressing mice. The new data shows that, similar to sWAT, ANG-2 overexpression significantly reduces the number of crown like structures and induces an increase in anti-inflammatory macrophage markers in eWAT. The new data are now added to Figure 4—figure supplement 1.

We further conducted the FACS analysis for pro- and anti- inflammatory macrophages (M1/M2) in both ANG-2 transgenic mice and ANG-2 antibody neutralizing mice. The results are in agreement with our previous histological and qPCR results. The new data are now added to Figure 4, Figure 4—figure supplement 2, Figure 6 and Figure 6—figure supplement 2.

*2) The data shows that ANG2 prevents diet-induced body weight gain. It would be exciting to test the therapeutic effect of ANG2; that is to determine if the inducible expression of ANG2 in obese mice sufficiently reverse obesity-linked glucose intolerance, inflammation, and lipid profile. If such studies have already been initiated and are well along, it would be great to include them. However, the reviewers recognize that starting such an experiment at this stage is beyond the scope of an eLife revision.*

Thanks for the reviewer’s question and the thoughtful suggestion.The suggested experiment is reasonable. We thus performed another animal study accordingly. Briefly, both control and ANG-2 transgenic mice were pre-induced by Dox for five weeks, and then we replaced the Dox food with a high-fat diet (HFD) and measured the body weights and OGTT on Day 0, Day 3, Day 7 and Day 10. We found that pre-induction of ANG-2 could prevent the mice from diet induced body weight gain and glucose intolerance, confirming the reviewer’s hypothesis. The new data are now added to Figure 3—figure supplement 2. The potential preventive role of ANG-2 induction against diet-induced obesity provides us further evidence that ANG-2 may serve as an important therapeutic target.

*3) Did ANG-2 overexpression reduce adipose tissue hypoxia (Figure 2)? The authors could examine HIF-1α protein and target gene expression in sWAT and eWAT. Pimonidazole staining would be helpful to access the degree of adipose tissue hypoxia upon ANG-2 overexpression.*

It is indeed important to assess the hypoxia in the adipose tissue when we observe increased vasculature in ANG-2 overexpressing mice. Following the suggestions, we checked the HIF-1α protein and mRNA levels in sWAT, and show a significant decrease of HIF-1α protein and gene expression. In parallel, we also performed a pimonidazole staining in sWAT from ANG-2 transgenic mice, and further validated that ANG-2 overexpression dramatically reduces pimonidazole positive signaling. Please see the new data in Figure 2—figure supplement 1.

*4) It would be insightful to determine if reduced adipose inflammation by ANG2 tg expression was associated with reduced adipose fibrosis.*

Thank you for the insightful suggestion. In the revision, we provided the trichrome staining images of sWAT sections and qPCR results for several fibrotic genes to show that there is a dramatic decrease of fibrosis in sWAT of ANG-2 transgenic mice. Please refer to the newly added Figure 4—figure supplement 3.

*5) The authors may wish to comment on the possibilities as to how the ANG2tg mice exhibited higher energy expenditure though ANG2 did not induce browning. Does ANG2 expression activate BAT thermogenesis?*

It is quite interesting that although ANG-2 overexpression induces vasculature, no browning effect is observed. When we carefully analyzed the brown adipose tissue (BAT) in ANG-2 transgenic mice, we found that both *Vegfa* and *Ucp1* gene expressions are elevated in BAT. Upon H&E staining of BAT sections, control mice show bigger lipid droplet accumulation and “whitening” phenotypes. In contrast, ANG-2 overexpressing mice maintain classical histological features of BAT. Thus, this suggests that ANG-2 overexpression could enhance BAT function under a HFD challenge. The new data are now added to Figure 4—figure supplement 4.

*6) Was insulin signaling cascade improved in adipose tissue of ANG-2 tg or ANG-2 neutralized mice (Figure 3 and Figure 7) upon HFD feeding?*

This is an excellent point. After we stimulated the mice with insulin injections, we could show that Akt phosphorylation is highly enhanced in ANG-2 transgenic mice; by contrast, the insulin signaling cascade is mildly suppressed in ANG-2 antibody neutralized mice. Please see the new data in Figure 3—figure supplement 1 and Figure 7—figure supplement 1.